# SCARA5 Is Overexpressed in Prostate Cancer and Linked to Poor Prognosis

**DOI:** 10.3390/diagnostics13132211

**Published:** 2023-06-29

**Authors:** Fidelis Andrea Flockerzi, Johannes Hohneck, Matthias Saar, Rainer Maria Bohle, Phillip Rolf Stahl

**Affiliations:** 1Department of Pathology, Saarland University Medical Center, 66424 Homburg, Germany; 2Department of Urology and Pediatric Urology, University Hospital, 52074 Aachen, Germany; 3Department of Urology and Pediatric Urology, Saarland University Medical Center, 66424 Homburg, Germany

**Keywords:** SCARA5, prostate cancer, THSD7A

## Abstract

Prostate cancer is one of the most common malignancies worldwide, showing a wide range of clinical behaviors. Therefore, several treatment options arise out of the diagnosis “prostate cancer”. For this reason, it is desirable to find novel prognostic and predictive markers. In former studies, we showed that THSD7A expression is associated with unfavorable prognostic parameters in prostate cancer and is linked to a high expression of focal adhesion kinase (FAK). Recently, scavenger receptor class A member 5 (SCARA5) was reported to be the downstream gene of THSD7A in esophageal squamous cell carcinoma. SCARA5 is believed to play an important role in the development and progression of several different tumor types. Most studies describe SCARA5 as a tumor suppressor. There is also evidence that SCARA 5 interacts with FAK. To examine the role of SCARA5 as a potential biomarker in prostate cancer, a total of 461 prostate cancers were analyzed via immunohistochemistry using tissue microarrays. Furthermore, we compared the expression level of SCARA5 with our previously collected data on THSD7A and FAK. High SCARA5 expression was associated with advanced tumor stage (*p* < 0.001), positive nodal status (*p* < 0.001) and high Gleason-score (*p* < 0.001). At least, strongly SCARA5-positive cancers were associated with THSD7A-positivity. There was no significant association between SCARA5 expression level and FAK expression level. To our knowledge, we are the first to investigate the role of SCARA5 in prostate cancer and we demonstrated that SCARA5 might be a potential biomarker in prostate cancer.

## 1. Introduction

Prostate cancer is one of the most common malignant tumors in men [1]. Most prostate cancers behave in an indolent manner. However, a given sub-group exhibits aggressive behavior, which potentially leads to systemic disease and death [2,3]. Because of this, several treatment options arise out of the diagnosis “prostate cancer”. These options range from watchful waiting and active surveillance to surgery, radiation, androgen-deprivation therapy and chemotherapy [4]. To date, the prostate-specific antigen (PSA)-level in blood, Gleason grade and tumor extension in biopsies remain the only established prognostic parameters in the preoperative setting. Given the wide range of therapeutic options and the concomitant side-effects, novel predictive (molecular) markers are desirable.

Previously, we showed that the expression of Thrombospondin type-1 domain-containing 7A (THSD7A) is associated with unfavorable prognostic parameters in prostate cancer, including tumor stage, Gleason grade and lymph node metastasis. THSD7A is also associated with early PSA recurrence [5,6]. THSD7A is hypothesized to be a membrane-associated N-glycoprotein with a soluble form produced by cells of endothelial and neuronal origin. The soluble form can promote endothelial cell migration, tube formation and sprouting in angiogenesis [7,8,9]. Several other studies indicate that THSD7A might play a role at least in the prognosis of different tumor types. Furthermore, there are several reports on the relationship between THSD7A expression in different tumor types and the development of a membranous nephropathy [10,11,12,13,14,15,16,17,18,19].

In addition, there is evidence that focal adhesion kinase (FAK) might be activated by THSD7A [7,8,9]. FAK is a protein tyrosine kinase that regulates cellular adhesion, motility, proliferation and survival in various types of cells, and meanwhile is quite an established tumor marker. FAK is overexpressed in several tumor types and is believed to play a role in tumor progression and metastasis [20,21,22,23]. Therefore, FAK is discussed as an effective cancer target in various tumors [24,25,26,27,28]. FAK, together with SRC and ETK, are the three major non-receptor tyrosine kinases forming the SRC tyrosine kinase complex which in prostate cancer is suggested to play an important role in the aberrant activation of the androgen receptor [29]. Furthermore, FAK activation is considered to be an important factor in androgen-independent progression to neuroendocrine carcinoma [30]. In a former study, we demonstrated that THSD7A-positivity is associated with a high expression of FAK in prostate cancer [6]. This finding might be proof of the actual involvement of THSD7A in FAK-dependent signaling pathways and might describe an independent method of tumor development. This is of special importance because THSD7A, as a membrane-associated protein, might serve as a putative therapeutic target in cancer therapy.

Recently, Jumai et al. reported scavenger receptor class A member 5 (SCARA5) to be the downstream gene of THSD7A in esophageal squamous cell carcinoma (ESCC) [31]. Not much is known about SCARA5. Scavenger receptors are a superfamily of membrane-bound receptors and there are five known members of scavenger receptor class A. These include type II transmembrane proteins that form homotrimers on the cell surface. Scavenger receptor class A members recognize various ligands and are involved in multiple biological pathways [32].

SCARA5 is also believed to play an important role in the development and progression of several different tumor types. In this connection, most studies describe SCARA5 as being a tumor suppressor, for example, in colorectal cancer, renal cell carcinoma, breast cancer, lung cancer, oral squamous cell carcinoma, hepatocellular carcinoma, osteosarcoma and gastric cancer [33,34,35,36,37,38,39,40,41,42,43,44,45].

Interestingly, there is also evidence that SCARA5 directly interacts with FAK. Wen et al. state that the overexpression of SCARA5 inhibits tumor proliferation and invasion in osteosarcoma and that the overexpression of SCARA5 significantly inhibits the phosphorylation and thus the activation of FAK in this tumor entity [43].

Huang et al. indicate that SCARA5 might act as a tumor suppressor in hepatocellular carcinoma (HCC) by preventing the phosphorylation of FAK and thereby inhibiting the activation of the FAK-Src-Cas pathway [45]. Yan et al. made a similar observation. They report on a significant reduction in the phosphorylation of FAK in human glioma cell lines when SCARA5 is upregulated [46].

Taken together, there is evidence that SCARA5 might be part of a potential THSD7A-driven method of tumor development, as well as being involved in FAK-dependent signaling pathways.

To our knowledge, no one has investigated the expression status of SCARA5 in prostate cancer so far. For this reason, no data exist on the association between SCARA5 expression and clinicopathological parameters in prostate cancer.

The aim of this study was to demonstrate that SCARA5 might be a potential biomarker in prostate cancer. The main objective was to examine the association between SCARA5 expression and common clinicopathological parameters. Furthermore, we were interested whether there is a correlation between SCARA5 expression and the expression level of THSD7A. As there might also be a connection between SCARA5 and FAK in cancer, we also wanted to correlate SCARA5 expression and the expression level of FAK in its unphosphorylated form.

Therefore, a previously described cohort of a total of 461 prostate cancers was analyzed for SCARA5 via immunohistochemistry (IHC), using tissue microarrays (TMAs). The results of this analysis were compared with our previously collected data on THSD7A and FAK.

## 2. Materials and Methods

### 2.1. Tissue Samples

Tissue samples from 461 primary surgically resected prostatectomy specimens were collected and were brought into a tissue microarray format. All patients received treatment at the Department of Urology at the Saarland University Medical Center, Saarland University, Homburg/Saar Germany. The treatment took place between 2012 and 2020. None of the patients received neoadjuvant therapy. This was an exclusion criterion. Detailed histopathological data on Gleason grade and pT-status were available for all patients; data on nodal status were lacking for six patients (Table 1).

### 2.2. Tissue Microarrays

Tissue microarrays were built as described previously [6]. In brief, a manual tissue arrayer was used and construction was performed following the manufacturer’s directions (Manual Tissue Arrayer, AlphaMetrix Biotech, Rödermark, Germany). Tissue cylinders were punched out of paraffin-embedded tissue blocks. These were previously evaluated with regard to their tumor content. Tissue cylinders were then brought into empty “recipient” paraffin blocks. Tissue cylinders had a diameter of 0.6 mm each.

### 2.3. Immunohistochemistry

To perform immunohistochemistry, 4 µm sections of the TMA blocks were transferred to adhesion slides (Matsunami TOMO, Osaka, Japan). For this purpose, a water bath (46 °C) was used and slides were dried overnight at 37 °C. Staining was performed using Benchmark Ultra (Ventana Medical Systems, Roche, Basel, Switzerland) and a primary antibody specific for SCARA5 (rabbit polyclonal antibody, abcam, Cambridge, UK; cat# ab118894; dilution 1:150) was utilized. Visualization of the bound antibody was performed using ultraView Universal Alkaline Phosphatase Red Detection (Roche, Basel, Switzerland) according to the manufacturer’s directions. CC1 buffer (Ventana, Roche, Basel, Switzerland) was used for heat-induced antigen retrieval at 95 °C for 64 min.

SCARA5 expression was evaluated by estimating the percentage of positive tumor cells. For each tissue sample, staining intensity was recorded semiquantitatively as 0, 1+, 2+, or 3+. For statistical analysis, the results were categorized into four groups: tumors with 1+ staining in ≤70% or with 2+ staining in ≤30% of cells were considered to be weakly positive. Tumors with 1+ staining in >70%, with 2+ staining in >30% but ≤70% and with 3+ staining in <30% of cells were considered to be moderately positive. Tumors with 2+ staining in >70% and with 3+ staining in ≥30% of cells were considered to be strongly positive. Tumors without any staining were considered to be negative. To our knowledge, no data exist on a validated scoring system for SCARA5 expression. To achieve a clear-cut evaluation, SCARA5 expression was dichotomized as being low-level (tumors with 0 staining, 1+ staining ≤70% and 2+ staining in ≤30%) and high-level (tumors with 1 + staining in >70%, 2+ staining in >30% of cells and 3+ staining in any tumor cell).

### 2.4. Statistics

Statistical analysis was performed using R (version 4.2.2, R Corporation 2021, R Foundation for Statistical Computing, Vienna, Austria). Pearson’s chi-squared test was used for testing the null hypothesis of independence. For cross tables larger than 2 × 2 row-wise, Fisher’s exact test and *p*-value adjustment via the Benjamini–Hochberg procedure were performed as post hoc analysis. *p*-values smaller than 0.05 were considered to be statistically significant.

## 3. Results

A total of 364 (79.0%) tumors were analyzable for SCARA5-IHC. A total of 97 (21.0%) tumors were not analyzable due to a total lack of tissue or due to a lack of unequivocal tumor tissue. A total of 178 (48.9%) tumors showed a low expression and a total of 186 (51.1%) tumors showed a high expression, with a cytoplasmic staining pattern, respectively. Non-tumor prostate tissue revealed low cytoplasmic SCARA5 expression. Representative images are shown in Figure 1a–d.

High-level SCARA5 expression was significantly associated with an advanced tumor stage (*p* < 0.001), positive nodal status (*p* < 0.001) and high Gleason-score (*p* < 0.001). The results are shown in detail in Table 1.

### 3.1. Association between SCARA5 Expression and THSD7A-Positivity

To evaluate a potential link between SCARA5 and THSD7A in prostate cancer, we compared the SCARA5 expression levels with our previously collected data on THSD7A. A total of 333 (72.2%) tumors were analyzable for SCARA5-IHC as well as for THSD7A-IHC. There was no significant association between SCARA5 expression level and THSD7A-positivity (*p* = 0.6) when utilizing our defined scoring criteria. The results are shown in Table 2.

However, focusing on tumors with strong SCARA5-positivity, we found an association. Strongly SCARA5-positive cancers were associated with THSD7A-positivity (8 of 35 analyzable samples, *p* = 0.016). The results are shown in Table 3.

### 3.2. Association between SCARA5 Expression and FAK Expression

To evaluate a potential link between SCARA5 and FAK in prostate cancer, we compared the SCARA5 expression levels with our previously collected data on FAK. A total of 305 (66.2%) tumors were analyzable for SCARA5-IHC as well as for FAK-IHC. There was no significant association between SCARA5 expression level and FAK expression level (*p* = 0.4). The results are shown in Table 4.

## 4. Discussion

Prostate cancer is one of the most common malignancies. However, the vast majority of patients with a diagnosis of prostate cancer will not die from the disease. Since a wide range of treatment options arise out of the diagnosis “prostate cancer”, some of them with concomitant side-effects, it is desirable to find novel prognostic and predictive (molecular) markers.

THSD7A is associated with unfavorable prognostic parameters in prostate cancer, as well as with early PSA recurrence. We also demonstrated that THSD7A-positivity is associated with high FAK expression in prostate cancer [5,6]. FAK is a protein tyrosine kinase and meanwhile is quite an established tumor marker which is overexpressed in several tumor types. FAK is believed to play a role in tumor progression and metastasis and is discussed as an effective cancer target in various tumors. This correlation might describe an independent method of tumor development.

Recently, Jumai et al. reported SCARA5 to be the downstream gene of THSD7A in esophageal squamous cell carcinoma. They also stated that SCARA5 promotes the proliferation and migration of ESCC cells and that SCARA5 is expressed differently between ESCC and normal esophageal tissue. However, they did not find any significant association between SCARA5 expression and clinicopathological parameters or prognosis [31]. There is also evidence that SCARA 5 directly interacts with FAK. Several groups state that SCARA5 overexpression inhibits the phosphorylation and thus the activation of FAK in different tumor types, for example osteosarcoma, hepatocellular carcinoma and glioma [43,45,46]. For this reason, we were interested in whether SCARA5 might play a role as a potential biomarker in prostate cancer. Furthermore, we wanted to find out whether there is an association between SCARA5 expression in this tumor entity and the expression status of THSD7A and FAK, respectively.

High-level SCARA5 expression was significantly associated with advanced tumor stage (*p* < 0.001), positive nodal status (*p* < 0.001) and high Gleason-score (*p* < 0.001) in prostate cancer.

Our results are not quite in the range of previously collected data. In contrast to our findings, the vast majority of studies describe SCARA5 as a protein with characteristics of a tumor suppressor gene. Several groups report on the downregulation of this protein, for example in colorectal cancer, renal cell carcinoma, breast cancer, oral squamous cell carcinoma, melanoma and gastric cancer [33,34,35,36,37,38,39,40,41,42,44,47,48,49].

Liu H et al. described that the low-level expression of SCARA5 was correlated with tumor stage, tumor size and venous metastasis in HCC. Additionally, they stated that patient survival was significantly better in the group with a higher expression of SCARA5 than that of the group with lower expression [48]. According to Liu J et al., SCARA5 mRNA levels are downregulated in colorectal cancer tissues compared with normal tissues and low SCARA5 expression is associated with poor prognosis in this tumor entity. They validated these results in clinical specimen using IHC [33]. Khamas et al. reported quite similar findings. They found down-expression of SCARA5 in colorectal cancer cell lines and in tumor tissue compared to non-tumor tissue [39]. Ulker et al. showed that SCARA5 expression was decreased in breast cancer tissue compared to non-tumor tissue. Furthermore, they demonstrated that SCARA5 expression was associated with histological grade [35]. You et al. also stated that SCARA5 is significantly downregulated in breast cancer tissues and cells and is correlated with tumor size, histological grade and lymph node metastasis. They also stated that an overexpression of SCARA5 suppressed cell proliferation and invasion [41]. Similar results were reported for gastric cancer. Zhang et al. showed that the protein level of SCARA5 was negatively associated with aggressive clinicopathological characteristics in this tumor entity, as well as with poor prognosis. SCARA5 overexpression markedly suppressed the growth, migration and invasion of gastric cancer cell lines in vitro and the upregulation of SCARA5 inhibited tumor growth and metastasis in a xenograft model [42]. Liu Y et al. indicate that SCARA5 expression is decreased in oral squamous cell carcinoma compared to normal oral mucosa and that the downregulation of SCARA5 is associated with cell proliferation and invasion [36]. Others showed that SCARA5 mRNA expression is significantly lower in melanoma than in adjacent normal skin. Ni et al. also state that the decreased expression of SCARA5 in melanoma is correlated with tumor stage, nodal status, metastasis and recurrence. Overall survival was significantly higher in tumors with high SCARA5 expression compared to tumors with low SCARA5 expression [49].

Our results suggest an opposite role of SCARA5 in prostate cancer. In this study, high expression levels of SCARA5 were significantly associated with adverse clinicopathological parameters. For this reason, SCARA5 has to be discussed as an oncogene rather than a tumor suppressor gene, at least in prostate cancer. Only one group reported similar results. Jumai et al. stated that SCARA5 is the downstream gene of THSD7A in esophageal squamous cell carcinoma and they showed that the expression levels of SCARA5 are significantly higher in tumor tissue compared to normal adjacent tissue.

In this context, we were also interested in a potential connection between THSD7A and SCARA5 in prostate cancer. Utilizing our defined scoring criteria, we could not find a significant association between SCARA5 expression levels and THSD7A-positivity. Notably, focusing on tumors with strong SCARA5-positivity, we found an association with THSD7A-positivity (8 of 35 analyzable samples, *p* = 0.016). Regarding these data, we cannot exclude a potential link between THSD7A and SCARA5.

As some authors had reported on the potential interaction of SCARA5 with FAK, we were interested in whether we could also demonstrate a correlation between the expression levels of these two proteins in prostate cancer. However, we were not able to find a significant association between SCARA5 expression levels and FAK expression levels, neither utilizing our defined scoring criteria, nor focusing on tumors with strong SCARA5-positivity.

Our work for sure has some limitations. Due to the retrospective nature of our study, we cannot provide data on PSA recurrence. For this reason, we cannot give a reliable statement on the correlation between SCARA5 expression and at least indirect survival rates. However, Gleason grade, tumor stage and nodal status are very strong prognostic indicators and there is a significant association between SCARA5 expression and these three strong prognostic indicators (*p* < 0.001, respectively). Furthermore, the analyzed sample size could be larger. This applies in particular regarding statistical analysis in prostate cancer, as one has to deal with a five-tiered grading system by the WHO. However, statistical analysis revealed significant associations between SCARA5 expression and clinicopathological parameters. Another limitation of this study is that we only used immunohistochemistry to determine the expression level of SCARA5. For this reason, we cannot provide reliable information on what causes the determined expression levels (for example, genetic alterations).

To our knowledge, we are the first to investigate the role of SCARA5 as a potential tumor marker in prostate cancer. We showed that high SCARA5 expression is associated with adverse clinicopathological parameters in this tumor entity. Although a very strong expression of SCARA5 seemed to be associated with THSD7A-positivity, we were not able to demonstrate a clear correlation between the expression of these two proteins. We also did not find a significant association of SCARA5 expression with FAK expression levels. In contrast to other tumor entities, SCARA5 shows characteristics of an oncogene in prostate cancer.

## 5. Conclusions

SCARA5 is overexpressed in prostate cancer and is significantly associated with adverse clinicopathological parameters. Therefore, SCARA5 is a potential tumor marker and tumor target in prostate cancer.

## Figures and Tables

**Figure 1 diagnostics-13-02211-f001:**
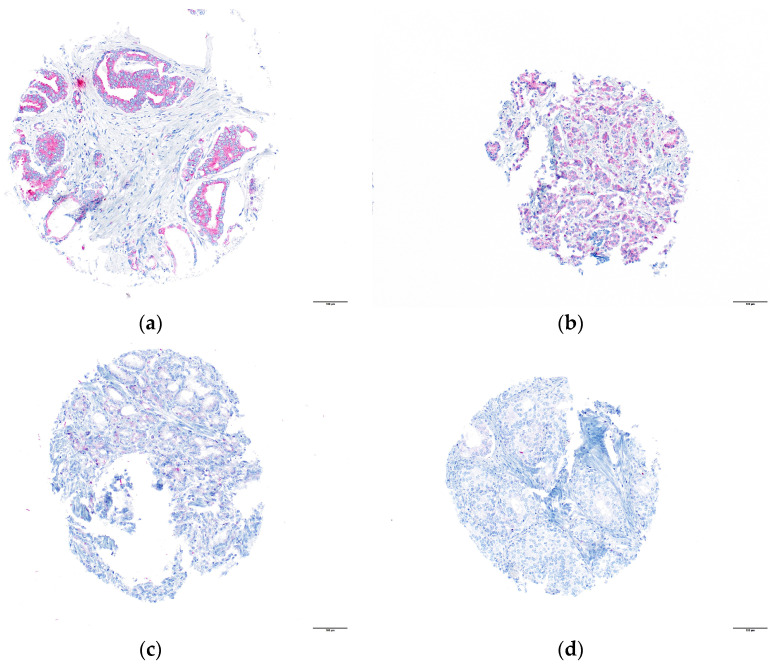
SCARA5 shows cytoplasmic staining pattern: (**a**) tumor cells with 3+ staining, (**b**) tumor cells with 2+ staining, (**c**) tumor cells with 1+ staining and (**d**) negative tumor tissue, magnification 100×.

**Table 1 diagnostics-13-02211-t001:** Association between SCARA5 expression and clinicopathological parameters.

	SCARA5		
	Low	High	*p*
pT-status			<0.001
pT2	109	67	
pT3–pT4	69	119	
Nodal status			<0.001
pN0	149	118	
pN+	26	65	
WHO grade group			<0.001
1	41	27	
2	56	28	
3	39	46	
4	37	74	
5	5	11	

**Table 2 diagnostics-13-02211-t002:** Association between SCARA5 expression and THSD7A-positivity.

	SCARA5		
	Low	High	*p*
THSD7A			0.6
negative	148	151	
positive	15	19	

**Table 3 diagnostics-13-02211-t003:** Association between SCARA5 expression and THSD7A-positivity with focus on the strong positive cases.

	SCARA5		
	Negative, Weak, Moderate	Strong	*p*
THSD7A			0.016
negative	272	27	
positive	26	8	

**Table 4 diagnostics-13-02211-t004:** Association between SCARA5 expression and FAK expression.

	SCARA5		
	Low	High	*p*
FAK			0.4
low	96	90	
high	55	64	

## Data Availability

The datasets used and analyzed in this paper are available from the corresponding author upon reasonable request.

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
