# Peer review of "SCARA5 Is Overexpressed in Prostate Cancer and Linked to Poor Prognosis"

_diagnostics, 2023, doi:10.3390/diagnostics13132211_

Round 1

Reviewer 1 Report

This article by Fidelis Andrea Flockerzi et al. describes the expression level of SCARA5 in prostate tumors. while the SCARA5 expression seems to be associated with prostate cancer progression, there are major flaws/drawbacks in the study as presented.

Overall, the manuscript is very poorly written and must be improved in a succinct manner; specifically, the abstract, introduction, and discussions.

Title, ‘SCARA5 is overexpressed in prostate cancer and linked to poor 2 prognosis’ does not justify the study results. There is no survival/prognostic analysis with respect to SCARA5 expression.

The abstract lacks the interpretation of the results for the SCARA5 expression in association with FAK and THSD7A and seems misleading.

Line #27: “SCARA5 is a potential biomarker,” Authors should lower the emphasis unless validated in a larger cohort with additional statistical analyses.

The inter-relationships between the SCARA5-THSD7A-FAK, and then FAK-ERK signaling are patchy and confusing in understanding the role of SCARA5 in prostate cancer.

The study is talking too much about their previous study on FAK and THSD7A, and the context of SCARA5 expression is diluted.

Line # 118: “Tissue microarrays were built as described previously (?). Should provide a reference.

The statement on the use of material for the research purpose with IRB approval should be included OR a reference should be provided on the use of biospecimens (as above; Line 118). 

The quality of staining in the provided monographs is poor.

The writing is difficult to understand; not necessarily the English language but the provided interpretation of the data seems cumbersome; questioning the English language interpretation. 

Author Response

We thank Reviewer 1 for his comments and tried to answer the requests in an appropriate way.

Title:

Title, ‘SCARA5 is overexpressed in prostate cancer and linked to poor prognosis’ does not justify the study results. There is no survival/prognostic analysis with respect to SCARA5 expression.

Reviewer 1 is of the opinion that the title of our manuscript is not justified by our study results because of the lack of a survival/prognostic analysis. In general, in studies dealing with prostate cancer statistics on overall survival are not provided. Reasons for that are a comparatively late onset of the disease and long overall survival. Many men die with prostate cancer but not as a cause of the disease. One possibility to counteract this problem is to check the correlation with PSA-recurrence. Due to the retrospective nature of our study we can not provide data on PSA-recurrence. However, Gleason grade, tumor stage and nodal status are very strong prognostic indicators. The significant association between SCARA5 expression and these three strong prognostic indicators (p < 0.001, respectively) for sure justifies the choice of words of our title.

Abstract

The abstract lacks the interpretation of the results for the SCARA5 expression in association with FAK and THSD7A and seems misleading.

Reviewer 1 asks us to add information on the association between SCARA5 expression with FAK and THSD7A, respectively. This information was added to the Abstract (Line 24 – 26 and Line 27 – 29).

Line #27: “SCARA5 is a potential biomarker,” Authors should lower the emphasis unless validated in a larger cohort with additional statistical analyses.

Reviewer 1 asks us to be a little bit more conservative regarding the meaning of SCARA5. We modified the above-mentioned sentence (Line 31)

The inter-relationships between the SCARA5-THSD7A-FAK, and then FAK-ERK signaling are patchy and confusing in understanding the role of SCARA5 in prostate cancer.

Reviewer 1 criticizes that the above-mentioned inter-relationships between SCARA5-THSD7A-FAK, and then FAK-ERK signaling are patchy and confusing. Therefore we removed the part concerning FAK-ERK singnaling (Line 90 – 91, Line 213-215). We do not think that the shown relationship between SCARA5-THSD7A-FAK is patchy. Till now, one can only find little data on this linkage. We tried to put the existing data in a reasonable context.

The study is talking too much about their previous study on FAK and THSD7A, and the context of SCARA5 expression is diluted.

Very less is known/published about SCARA5. To our knowledge, we are the first to present data on SCARA5 in prostate cancer. There are a few studies that deal with the relationship between FAK and THSD7A with SCARA5, respectively. We think it is important to provide sufficient data on a possible relationship between SCARA5 and other, better known, proteins to find possible interpretations what kind of role SCARA5 might play in prostate cancer. Especially, if one is dealing with a new potential biomarker it is necessary to find a linkage to known and better described pathways. In our opinion, providing background on other proteins which might be connected to SCARA5 does not dilute the context of SCARA5 expression, but rather emphasizes a potential role for SCARA5 in cancer. Aside from that, we limited the information of our previous work to the Introduction. Discussion mainly deals with SCARA5.

Line # 118: “Tissue microarrays were built as described previously (?). Should provide a reference.

Reviewer 1 asks us to provide a reference for the above-mentioned sentence. We added this reference (Line 120)

The statement on the use of material for the research purpose with IRB approval should be included OR a reference should be provided on the use of biospecimens (as above; Line 118). 

This statement is provided at the end of the manuscript (Line 291)

Institutional Review Board Statement: The study was conducted in accordance with the Declaration of Helsinki. The study has been approved by the ethics committee of the medical chamber of the Saarland, Germany (Identification number 282/19).

The quality of staining in the provided monographs is poor.

 Reviewer 1 states that the quality of staining in the provided monographs is poor. We are convinced that the immunohistochemical staining quality is quite good. We could easily discriminate different staining intensities. With the utilized protocol for immunihistochemistry we were able to show significant associations.

Reviewer 2 Report

Authors should be congratulated for their work. The topic is interesting and intriguing. The role of the genetic background that characterizes the different types of Prostate cancer represents an hot topic to address. The role of SCARA5 was recorded for several other malignancies (breast, stomach, colorectal, bladder). The tables and figures are of good quality. There are minor typos in the manuscript and the various abbreviations should be defined in the correct order (FAK was never defined in the abstract, SCARA5 was defined and then the authors used the extensive form; please, be consistent)

A minor revision is required

Author Response

We thank Reviewer 2 for his comments and tried to answer the requests in an appropriate way.

Authors should be congratulated for their work. The topic is interesting and intriguing. The role of the genetic background that characterizes the different types of Prostate cancer represents an hot topic to address. The role of SCARA5 was recorded for several other malignancies (breast, stomach, colorectal, bladder). The tables and figures are of good quality. There are minor typos in the manuscript and the various abbreviations should be defined in the correct order (FAK was never defined in the abstract, SCARA5 was defined and then the authors used the extensive form; please, be consistent)

A minor revision is required

Reviewer 2 asks us to define the used abbreviations in the correct order. We improved these irregularities (Line 17-18, Line 72, Line 205, Line 259-260, Line 267)

We also improved some small mistakes (Line 84, Line 116, Line 176, Line 194-195, Line 255, Line 267, Line 268, Line 282)